# ViTGAN: Training GANs with Vision Transformers

**Kwonjoon Lee**[1,3]  **Huiwen Chang**[2]  **Lu Jiang**[2]  **Han Zhang**[2]  **Zhuowen Tu**[1]  **Ce Liu**[4]

[1]UC San Diego  [2]Google Research  [3]Honda Research Institute  [4]Microsoft Azure AI

kwl042@eng.ucsd.edu  {huiwenchang,lujiang,zhanghan}@google.com

ztu@ucsd.edu  ce.liu@microsoft.com

## Abstract

Recently, Vision Transformers (ViTs) have shown competitive performance on image recognition while requiring less vision-specific inductive biases. In this paper, we investigate if such performance can be extended to image generation. To this end, we integrate the ViT architecture into generative adversarial networks (GANs). For ViT discriminators, we observe that existing regularization methods for GANs interact poorly with self-attention, causing serious instability during training. To resolve this issue, we introduce several novel regularization techniques for training GANs with ViTs. For ViT generators, we examine architectural choices for latent and pixel mapping layers to facilitate convergence. Empirically, our approach, named *ViTGAN*, achieves comparable performance to the leading CNN-based GAN models on three datasets: CIFAR-10, CelebA, and LSUN bedroom. Our code is available online[1].

## 1 Introduction

Convolutional neural networks (CNNs) (LeCun et al., 1989) are dominating computer vision today, thanks to their powerful capability of convolution (weight-sharing and local-connectivity) and pooling (translation equivariance). Recently, however, Transformer architectures (Vaswani et al., 2017) have started to rival CNNs in many vision tasks.

In particular, Vision Transformers (ViTs) (Dosovitskiy et al., 2021), which interpret an image as a sequence of *tokens* (analogous to *words* in natural language), have been shown to achieve comparable classification accuracy with smaller computational budgets (*i.e.*, fewer FLOPs) on the ImageNet benchmark. Unlike CNNs, ViTs capture a different inductive bias through self-attention where each patch is attended to *all* patches of the same image. ViTs, along with their variants (Touvron et al., 2020; Tolstikhin et al., 2021), though still in their infancy, have demonstrated advantages in modeling non-local contextual dependencies (Ranftl et al., 2021; Strudel et al., 2021) as well as promising efficiency and scalability. Since their recent inception, ViTs have been used in various tasks such as object detection (Beal et al., 2020), video recognition (Bertasius et al., 2021; Arnab et al., 2021), multitask pre-training (Chen et al., 2020a), *etc*.

In this paper, we examine whether Vision Transformers can perform the task of image generation *without using convolution or pooling*, and more specifically, whether ViTs can be used to train generative adversarial networks (GANs) with comparable quality to CNN-based GANs. While we can naively train GANs following the design of the standard ViT (Dosovitskiy et al., 2021), we find that GAN training becomes highly unstable when coupled with ViTs, and that adversarial training is frequently hindered by high-variance gradients in the later stage of discriminator training. Furthermore, conventional regularization methods such as gradient penalty (Gulrajani et al., 2017; Mescheder et al., 2018), spectral normalization (Miyato et al., 2018) cannot resolve the instability issue, even though they are proved to be effective for CNN-based GAN models (shown in Fig. 4). As unstable training is uncommon in the CNN-based GANs training with appropriate regularization, this presents a unique challenge to the design of ViT-based GANs.

---

[1]https://github.com/mlpc-ucsd/ViTGAN

We propose several necessary modifications to stabilize the training dynamics and facilitate the convergence of ViT-based GANs. In the *discriminator*, we design an improved spectral normalization that enforces Lipschitz continuity for stabilizing the training dynamics. In the *generator*, we propose two key modifications to the layer normalization and output mapping layers after studying several architecture designs. Our ablation experiments validate the necessity of the proposed techniques and their central role in achieving stable and superior image generation.

The experiments are conducted on three public image synthesis benchmarks: CIFAR-10, CelebA, and LSUN bedroom. The results show that our model, named *ViTGAN*, yields comparable performance to the leading CNN-based StyleGAN2 (Karras et al., 2020b; Zhao et al., 2020a) when trained under the same setting. Moreover, we are able to outperform StyleGAN2 by combining the StyleGAN2 discriminator with our ViTGAN generator.

Note that it is not our intention to claim ViTGAN is superior to the best-performing GAN models such as StyleGAN2 + ADA (Karras et al., 2020a) which are equipped with highly-optimized hyper-parameters, architecture configurations, and sophisticated data augmentation methods. Instead, our work aims to close the performance gap between the conventional CNN-based GAN architectures and the novel GAN architecture composed of vanilla ViT layers. Furthermore, the ablation in Table 4 shows the advantages of ViT's intrinsic capability (*i.e.*, adaptive connection weight and global context) for image generation.

## 2    RELATED WORK

**Generative Adversarial Networks**    Generative adversarial networks (GANs) (Goodfellow et al., 2014) model the target distribution using adversarial learning. It is typically formulated as a min-max optimization problem minimizing some distance between the real and generated data distributions, *e.g.*, through various $f$-divergences (Nowozin et al., 2016) or integral probability metrics (IPMs) (Müller, 1997; Song & Ermon, 2020) such as the Wasserstein distance (Arjovsky et al., 2017).

GAN models are notorious for unstable training dynamics. As a result, numerous efforts have been proposed to stabilize training, thereby ensuring convergence. Common approaches include spectral normalization (Miyato et al., 2018), gradient penalty (Gulrajani et al., 2017; Mescheder et al., 2018; Kodali et al., 2017), consistency regularization (Zhang et al., 2020; Zhao et al., 2021), and data augmentation (Zhao et al., 2020a; Karras et al., 2020a; Zhao et al., 2020b; Tran et al., 2021). These techniques are all designed inside convolutional neural networks (CNN) and have been only verified in convolutional GAN models. However, we find that these methods are insufficient for stabilizing the training of Transformer-based GANs. A similar finding was reported in (Chen et al., 2021) on a different task of pretraining. This paper introduces several novel techniques to overcome the unstable adversarial training of Vision Transformers.

**Vision Transformers**    Vision Transformer (ViT) (Dosovitskiy et al., 2021) is a convolution-free Transformer that performs image classification over a sequence of image patches. ViT demonstrates the superiority of the Transformer architecture over the classical CNNs by taking advantage of pretraining on large-scale datasets. Afterward, DeiT (Touvron et al., 2020) improves ViTs' sample efficiency using knowledge distillation as well as regularization tricks. MLP-Mixer (Tolstikhin et al., 2021) further drops self-attention and replaces it with an MLP to mix the per-location feature. In parallel, ViT has been extended to various computer vision tasks such as object detection (Beal et al., 2020), action recognition in video (Bertasius et al., 2021; Arnab et al., 2021), and multitask pretraining (Chen et al., 2020a). Our work is among the first to exploit Vision Transformers in the GAN model for image generation.

**Generative Transformers in Vision**    Motivated by the success of GPT-3 (Brown et al., 2020), a few pilot works study image generation using Transformer by autoregressive learning (Chen et al., 2020b; Esser et al., 2021) or cross-modal learning between image and text (Ramesh et al., 2021). These methods are different from ours as they model image generation as a autoregressive sequence learning problem. On the contrary, our work trains Vision Transformers in the generative adversarial training paradigm. Recent work of (Hudson & Zitnick, 2021), embeds (cross-)attention module within the CNN backbone (Karras et al., 2020b) in a similar spirit to (Zhang et al., 2019). The closest work to ours is TransGAN (Jiang et al., 2021), presenting a GAN model based on Swin Transformer

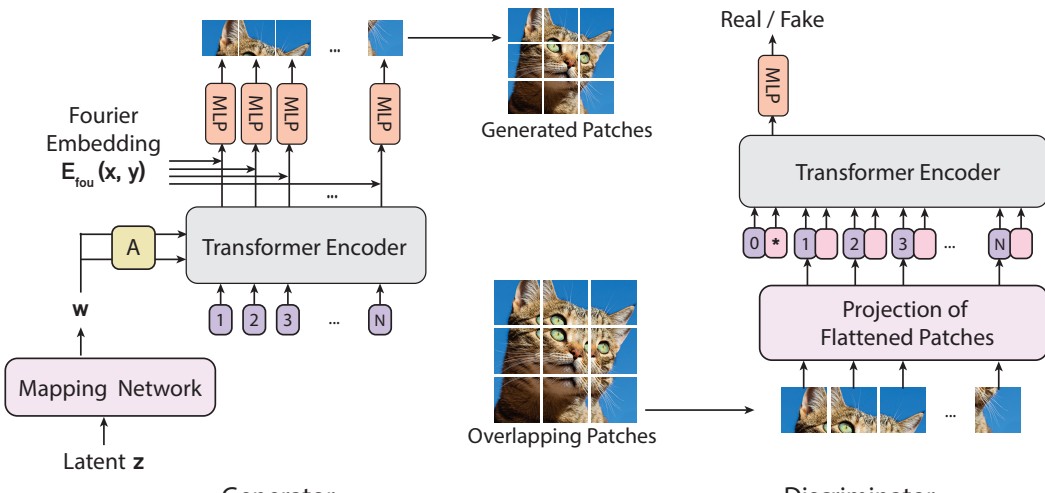

Figure 1: **Overview of the proposed ViTGAN framework.** Both the generator and the discriminator are designed based on the Vision Transformer (ViT). Discriminator score is derived from the classification embedding (denoted as [*] in the Figure). The generator generates pixels patch-by-patch based on patch embeddings.

backbone (Liu et al., 2021b). Our approach is complementary to theirs as we propose key techniques for training stability within the original ViT backbone (Dosovitskiy et al., 2021).

## 3 PRELIMINARIES: VISION TRANSFORMERS (VITS)

Vision Transformer (Dosovitskiy et al., 2021) is a pure transformer architecture for image classification that operates upon a sequence of image patches. The 2D image $\mathbf{x} \in \mathbb{R}^{H \times W \times C}$ is flattened into a sequence of image patches, following the raster scan, denoted by $\mathbf{x}_p \in \mathbb{R}^{N \times (P^2 \cdot C)}$, where $N = \frac{H \times W}{P^2}$ is the effective sequence length and $P \times P \times C$ is the dimension of each image patch.

Following BERT (Devlin et al., 2019), a learnable classification embedding $\mathbf{x}_{\text{class}}$ is prepended to the image sequence along with the added 1D positional embeddings $\mathbf{E}_{pos}$ to formulate the patch embedding $\mathbf{h}_0$. The architecture of ViT follows the Transformer architecture (Vaswani et al., 2017).

$$\mathbf{h}_0 = [\mathbf{x}_{\text{class}}; \mathbf{x}_p^1 \mathbf{E}; \mathbf{x}_p^2 \mathbf{E}; \cdots; \mathbf{x}_p^N \mathbf{E}] + \mathbf{E}_{pos}, \qquad \mathbf{E} \in \mathbb{R}^{(P^2 \cdot C) \times D}, \mathbf{E}_{pos} \in \mathbb{R}^{(N+1) \times D} \qquad (1)$$

$$\mathbf{h}'_\ell = \text{MSA}(\text{LN}(\mathbf{h}_{\ell-1})) + \mathbf{h}_{\ell-1}, \qquad \ell = 1, \ldots, L \qquad (2)$$

$$\mathbf{h}_\ell = \text{MLP}(\text{LN}(\mathbf{h}'_\ell)) + \mathbf{h}'_\ell, \qquad \ell = 1, \ldots, L \qquad (3)$$

$$\mathbf{y} = \text{LN}(\mathbf{h}_L^0) \qquad (4)$$

Equation 2 applies multi-headed self-attention (MSA). Given learnable matrices $\mathbf{W}_q, \mathbf{W}_k, \mathbf{W}_v$ corresponding to query, key, and value representations, a single self-attention head is computed by:

$$\text{Attention}_h(\mathbf{X}) = \text{softmax}\left(\frac{\mathbf{Q}\mathbf{K}^\top}{\sqrt{d_h}}\right)\mathbf{V}, \qquad (5)$$

where $\mathbf{Q} = \mathbf{X}\mathbf{W}_q$, $\mathbf{K} = \mathbf{X}\mathbf{W}_k$, and $\mathbf{V} = \mathbf{X}\mathbf{W}_v$. Multi-headed self-attention aggregates information from $H$ self-attention heads by means of concatenation and linear projection: $\text{MSA}(\mathbf{X}) = \text{concat}_{h=1}^H [\text{Attention}_h(\mathbf{X})]\mathbf{W} + \mathbf{b}$.

## 4 METHOD

Fig. 1 illustrates the architecture of the proposed ViTGAN with a ViT discriminator and a ViT-based generator. We find that directly using ViT as the discriminator makes the training volatile. We introduce techniques to both generator and discriminator to stabilize the training dynamics and facilitate the convergence: *(1)* regularization on ViT discriminator and *(2)* new architecture for generator.

### 4.1 REGULARIZING ViT-BASED DISCRIMINATOR

**Enforcing Lipschitzness of Transformer Discriminator**   Lipschitz continuity plays a critical role in GAN discriminators. It was first brought to attention as a condition to approximate the Wasserstein distance in WGAN (Arjovsky et al., 2017), and later was confirmed in other GAN settings (Fedus et al., 2018; Miyato et al., 2018; Zhang et al., 2019) beyond the Wasserstein loss. In particular, (Zhou et al., 2019) proves that Lipschitz discriminator guarantees the existence of the optimal discriminative function as well as the existence of a unique Nash equilibrium. A very recent work (Kim et al., 2021), however, shows that Lipschitz constant of standard dot product self-attention (*i.e.*, Equation 5) layer can be unbounded, rendering Lipschitz continuity violated in ViTs. To enforce Lipschitzness of our ViT discriminator, we adopt *L2 attention* proposed in (Kim et al., 2021). As shown in Equation 6, we replace the dot product similarity with Euclidean distance and also tie the weights for the projection matrices for query and key in self-attention:

$$\text{Attention}_h(\mathbf{X}) = \text{softmax}\Big(-\frac{d(\mathbf{X}\mathbf{W}_q, \mathbf{X}\mathbf{W}_k)}{\sqrt{d_h}}\Big)\mathbf{X}\mathbf{W}_v, \quad \text{where} \quad \mathbf{W}_q = \mathbf{W}_k, \tag{6}$$

$\mathbf{W}_q$, $\mathbf{W}_k$, and $\mathbf{W}_v$ are the projection matrices for query, key, and value, respectively. $d(\cdot, \cdot)$ computes *vectorized L2 distances* between two sets of points. $\sqrt{d_h}$ is the feature dimension for each head. This modification improves the stability of Transformers when used for GAN discriminators.

**Improved Spectral Normalization.**   To further strengthen the Lipschitz continuity, we also apply spectral normalization (SN) (Miyato et al., 2018) in the discriminator training. The standard SN uses power iterations to estimate spectral norm of the projection matrix for each layer in the neural network. Then it divides the weight matrix with the estimated spectral norm, so Lipschitz constant of the resulting projection matrix equals 1. We observe that making the spectral norm equal to 1 stabilized the training but GANs seemed to be underfitting in such settings (*c.f.*Table 3b). Similarly, we find R1 gradient penalty cripples GAN training when ViT-based discriminators are used (*c.f.*Figure 4). (Dong et al., 2021) suggests that the small Lipschitz constant of MLP block may cause the output of Transformer collapse to a rank-1 matrix. Without the spectral normalization, our GAN model **initially** learns in a healthy manner but the training becomes unstable (later on) since we cannot guarantee Lipschitzness of Transformer discriminator.

We find that multiplying the normalized weight matrix of each layer with the **spectral norm at initialization** is sufficient to solve this problem. Concretely, we use the following update rule for our spectral normalization, where $\sigma$ computes the standard spectral norm of weight matrices:

$$\bar{W}_{\text{ISN}}(\mathbf{W}) := \sigma(\mathbf{W}_{init}) \cdot \mathbf{W}/\sigma(\mathbf{W}). \tag{7}$$

**Overlapping Image Patches.**   ViT discriminators are prone to overfitting due to their exceeding learning capacity. Our discriminator and generator use the same image representation that partitions an image as a sequence of non-overlapping patches according to a predefined grid $P \times P$. These arbitrary grid partitions, if not carefully tuned, may encourage the discriminator to memorize local cues and stop providing meaningful loss for the generator. We use a simple trick to mitigate this issue by allowing some overlap between image patches (Liu et al., 2021b;a). For each border edge of the patch, we extend it by $o$ pixels, making the effective patch size $(P + 2o) \times (P + 2o)$.

This results in a sequence with the same length but less sensitivity to the predefined grids. It may also give the Transformer a better sense of which ones are neighboring patches to the current patch, hence giving a better sense of locality.

**Convolutional Projection.**   To allow Transformers to leverage local context as well as global context, we apply convolutions when computing $\mathbf{Q}, \mathbf{K}, \mathbf{V}$ in Equation 5. While variants of this idea were proposed in (Wu et al., 2021; Guo et al., 2021), we find the following simple option works well: we apply 3×3 convolution after reshaping image token embeddings into a feature map of size $\frac{H}{P} \times \frac{W}{P} \times D$. We note that this formulation does not harm the expressiveness of the original Transformer, as the original $\mathbf{Q}, \mathbf{K}, \mathbf{V}$ projection can be recovered by using the identity convolution kernel.

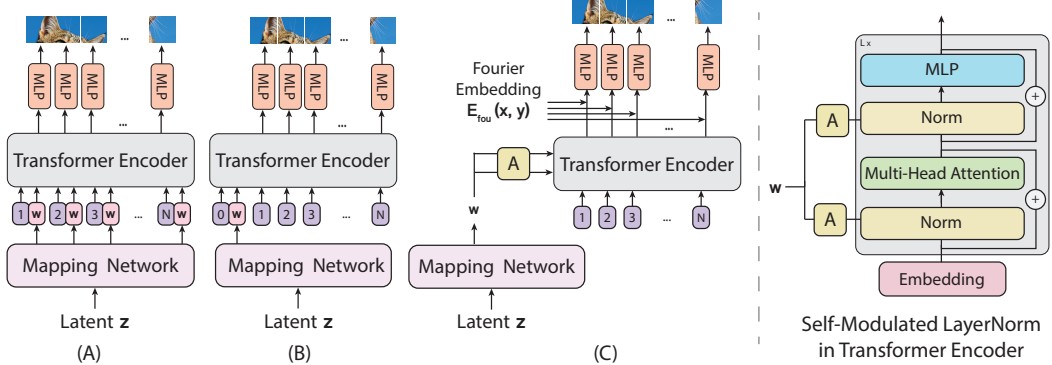

Figure 2: **Generator Architecture Variants.** The diagram on the left shows three generator architectures we consider: **(A)** adding intermediate latent embedding **w** to every positional embedding, **(B)** prepending **w** to the sequence, and **(C)** replacing normalization with self-modulated layernorm (SLN) computed by learned affine transform (denoted as **A** in the figure) from **w**. On the right, we show the details of the self-modulation operation applied in the Transformer block.

## 4.2 GENERATOR DESIGN

Designing a generator based on the ViT architecture is a nontrivial task. A challenge is converting ViT from predicting a set of class labels to generating pixels over a spatial region. Before introducing our model, we discuss two plausible baseline models, as shown in Fig. 2 (A) and 2 (B). Both models swap ViT's input and output to generate pixels from embeddings, specifically from the latent vector **w** derived from a Gaussian noise vector **z** by an MLP, *i.e.*, $\mathbf{w} = \text{MLP}(\mathbf{z})$ (called mapping network (Karras et al., 2019) in Fig. 2). The two baseline generators differ in their input sequences. Fig. 2 (A) takes as input a sequence of positional embeddings and adds the intermediate latent vector **w** to every positional embedding. Alternatively, Fig. 2 (B) prepends the sequence with the latent vector. This design is inspired by inverting ViT where **w** is used to replace the classification embedding $\mathbf{h}_L^0$ in Equation 4.

To generate pixel values, a linear projection $\mathbf{E} \in \mathbb{R}^{D \times (P^2 \cdot C)}$ is learned in both models to map a $D$-dimensional output embedding to an image patch of shape $P \times P \times C$. The sequence of $N = \frac{H \times W}{P^2}$ image patches $[\mathbf{x}_p^i]_{i=1}^N$ are finally reshaped to form an whole image **x**.

These baseline transformers perform poorly compared to the CNN-based generator. We propose a novel generator following the design principle of ViT. Our ViTGAN Generator, shown in Fig. 2 (c), consists of two components (1) a Transformer block and (2) an output mapping layer.

$$\mathbf{h}_0 = \mathbf{E}_{pos}, \qquad\qquad \mathbf{E}_{pos} \in \mathbb{R}^{N \times D}, \qquad (8)$$

$$\mathbf{h}'_\ell = \text{MSA}(\text{SLN}(\mathbf{h}_{\ell-1}, \mathbf{w})) + \mathbf{h}_{\ell-1}, \qquad \ell = 1, \ldots, L, \mathbf{w} \in \mathbb{R}^D \qquad (9)$$

$$\mathbf{h}_\ell = \text{MLP}(\text{SLN}(\mathbf{h}'_\ell, \mathbf{w})) + \mathbf{h}'_\ell, \qquad \ell = 1, \ldots, L \qquad (10)$$

$$\mathbf{y} = \text{SLN}(\mathbf{h}_L, \mathbf{w}) = [\mathbf{y}^1, \cdots, \mathbf{y}^N] \qquad \mathbf{y}^1, \ldots, \mathbf{y}^N \in \mathbb{R}^D \qquad (11)$$

$$\mathbf{x} = [\mathbf{x}_p^1, \cdots, \mathbf{x}_p^N] = [f_\theta(\mathbf{E}_{fou}, \mathbf{y}^1), \ldots, f_\theta(\mathbf{E}_{fou}, \mathbf{y}^N)] \quad \mathbf{x}_p^i \in \mathbb{R}^{P^2 \times C}, \mathbf{x} \in \mathbb{R}^{H \times W \times C} \quad (12)$$

The proposed generator incorporates two modifications to facilitate the training.

**Self-Modulated LayerNorm.** Instead of sending the noise vector **z** as the input to ViT, we use **z** to modulate the layernorm operation in Equation 9. This is known as self-modulation (Chen et al., 2019) since the modulation depends on no external information. The self-modulated layernorm (SLN) in Equation 9 is computed by:

$$\text{SLN}(\mathbf{h}_\ell, \mathbf{w}) = \text{SLN}(\mathbf{h}_\ell, \text{MLP}(\mathbf{z})) = \gamma_\ell(\mathbf{w}) \odot \frac{\mathbf{h}_\ell - \boldsymbol{\mu}}{\boldsymbol{\sigma}} + \beta_\ell(\mathbf{w}), \qquad (13)$$

where $\boldsymbol{\mu}$ and $\boldsymbol{\sigma}$ track the mean and the variance of the summed inputs within the layer, and $\gamma_l$ and $\beta_l$ compute adaptive normalization parameters controlled by the latent vector derived from **z**. $\odot$ is the element-wise dot product.

Table 1: **Comparison to representative GAN architectures on unconditional image generation benchmarks.** *Results from the original papers. All other results are our replications and trained with DiffAug (Zhao et al., 2020a) + bCR for fair comparison. ↓ means lower is better.

| Architecture | CIFAR 10 | | CelebA 64x64 | | LSUN 64x64 | | LSUN 128x128 | |
|---|---|---|---|---|---|---|---|---|
| | FID ↓ | IS ↑ | FID ↓ | IS ↑ | FID ↓ | IS ↑ | FID ↓ | IS ↑ |
| BigGAN+ DiffAug (Zhao et al., 2020a) | 8.59* | 9.25* | - | - | - | - | - | - |
| StyleGAN2 (Karras et al., 2020b) | 5.60 | 9.41 | **3.39** | **3.43** | 2.33 | 2.44 | 3.26 | 2.26 |
| TransGAN (Jiang et al., 2021) | 9.02* | 9.26* | - | - | - | - | - | - |
| Vanilla-ViT | 12.7 | 8.40 | 20.2 | 2.57 | 218.1 | 2.20 | - | - |
| ViTGAN (Ours) | 4.92 | 9.69 | 3.74 | 3.21 | 2.40 | 2.28 | 2.48 | 2.26 |
| StyleGAN2-D+ViTGAN-G (Ours) | **4.57** | **9.89** | - | - | **1.49** | **2.46** | **1.87** | **2.32** |

**Implicit Neural Representation for Patch Generation.** We use an implicit neural representation (Park et al., 2019; Mescheder et al., 2019; Tancik et al., 2020; Sitzmann et al., 2020) to learn a continuous mapping from a patch embedding $\mathbf{y}^i \in \mathbb{R}^D$ to patch pixel values $\mathbf{x}_p^i \in \mathbb{R}^{P^2 \times C}$. When coupled with Fourier features (Tancik et al., 2020) or sinusoidal activation functions (Sitzmann et al., 2020), implicit representations can constrain the space of generated samples to the space of smooth-varying natural signals. Concretely, similarly to (Anokhin et al., 2021), $\mathbf{x}_p^i = f_\theta(\mathbf{E}_{fou}, \mathbf{y}^i)$ where $\mathbf{E}_{fou} \in \mathbb{R}^{P^2 \cdot D}$ is a Fourier encoding of $P \times P$ spatial locations and $f_\theta(\cdot, \cdot)$ is a 2-layer MLP. For details, please refer to Appendix D. We find implicit representation to be particularly helpful for training GANs with ViT-based generators, *c.f.* Table 3a.

## 5 EXPERIMENTS

We train and evaluate our model on various standard benchmarks for image generation, including *CIFAR-10* (Krizhevsky et al., 2009), *LSUN bedroom* (Yu et al., 2015) and *CelebA* (Liu et al., 2015). We consider three resolution settings: 32×32, 64×64, 128×128, and 256×256. We strive to use a consistent setup across different resolutions and datasets, and as such, keep all key hyper-parameters, except for the number of Transformer blocks and patch sizes, the same. For more details about the datasets and implementation, please refer to Appendix C and D.

### 5.1 MAIN RESULTS

Table 1 shows the main results on three standard benchmarks for image synthesis. Our method is compared with the following baseline architectures. *TransGAN* (Jiang et al., 2021) is the only existing convolution-free GAN that is entirely built on the Transformer architecture. *Vanilla-ViT* is a ViT-based GAN that employs the generator illustrated in Fig. 2 (A) and a vanilla ViT discriminator without any techniques discussed in Section 4.1. For a fair comparison, R1 penalty and bCR (Zhao et al., 2021) + DiffAug (Zhao et al., 2020a) were used for this baseline. The architecture with the generator illustrated in Fig. 2 (B) is separately compared in Table 3a. In addition, BigGAN (Brock et al., 2019) and StyleGAN2 (Karras et al., 2020b) are also included as state-of-the-art CNN-based GAN models.

Our ViTGAN model outperforms other Transformer-based GAN models by a large margin. This stems from the improved stable GAN training on the Transformer architecture. As shown in Fig. 4, our method overcomes the spikes in gradient magnitude and stabilizes the training dynamics. This enables ViTGAN to converge to either a lower or a comparable FID on different datasets.

ViTGAN achieves comparable performance to the leading CNN-based models, *i.e.* BigGAN and StyleGAN2. Note that in Table 1, to focus on comparing the architectures, we use a generic version of StyleGAN2. More comprehensive comparisons with StyleGAN2 are included in Appendix A.1. As shown in Fig. 3, the image fidelity of the best Transformer baseline (Middle Row) has been notably improved by the proposed ViTGAN model (Last Row). Even compared with StyleGAN2, ViTGAN generates images with comparable quality and diversity. Notice there appears to be a perceivable difference between the images generated by Transformers and CNNs, *e.g.* in the background of the CelebA images. Both the quantitative results and qualitative comparison substantiate the efficacy of the proposed ViTGAN as a competitive Transformer-based GAN model.

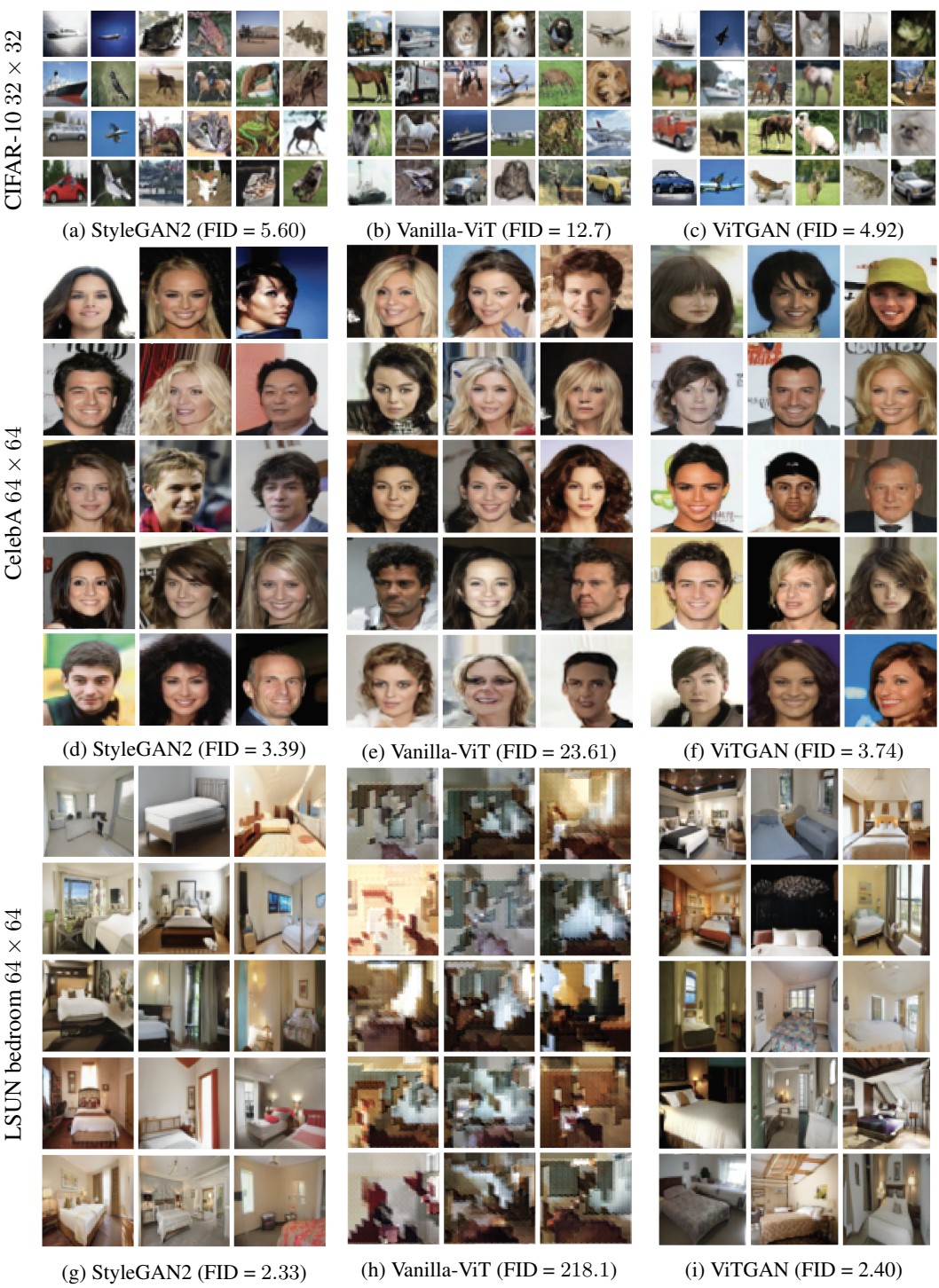

Figure 3: **Qualitative Comparison.** We compare our ViTGAN with StyleGAN2, and our best Transformer baseline, *i.e.*, a vanilla pair of ViT generator and discriminator described in Section 5.1, on the CIFAR-10 $32 \times 32$, CelebA $64 \times 64$ and LSUN Bedroom $64 \times 64$ datasets. Results on LSUN Bedroom $128 \times 128$ and $256 \times 256$ can be found in the Appendix.

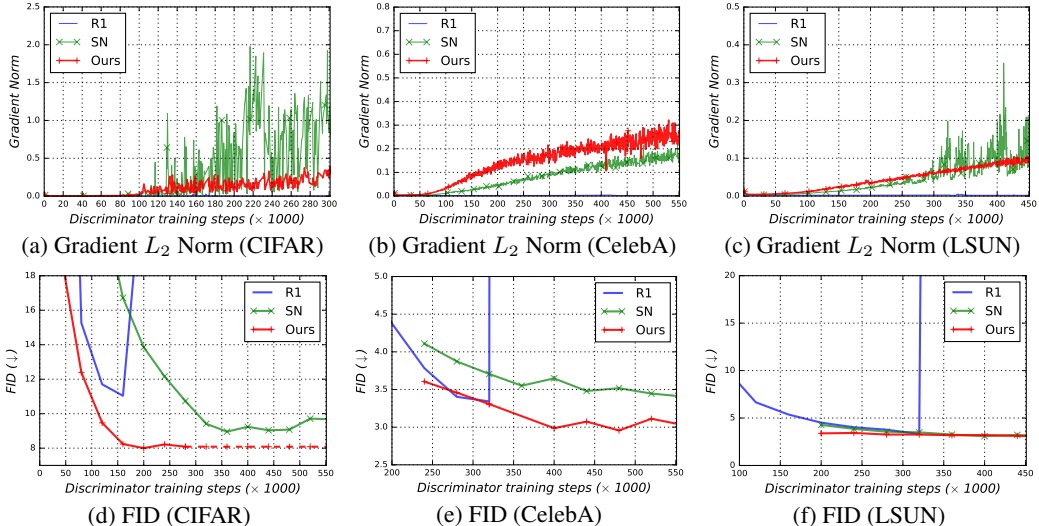

Figure 4: **(a-c) Gradient magnitude ($L_2$ norm over all parameters) of ViT discriminator and (d-f) FID score (lower the better)** as a function of training iteration. Our ViTGAN are compared with two baselines of Vanilla ViT discriminators with R1 penalty and spectral norm (SN). The remaining architectures are the same for all methods. Our method overcomes the spikes of gradient magnitude, and achieve lower FIDs (on CIFAR and CelebA) or a comparable FID (on LSUN).

## 5.2 ABLATION STUDIES

We conduct ablation experiments on the CIFAR dataset to study the contributions of the key techniques and verify the design choices in our model.

**Compatibility with CNN-based GANs**  In Table 2, we mix and match the generator and discriminator of our ViTGAN and the leading CNN-based GAN: StyleGAN2. With the Style-GAN2 generator, our ViTGAN discriminator outperforms the vanilla ViT discriminator. Besides, our ViTGAN generator still works together with the StyleGAN2 discriminator. The results show the proposed techniques are compatible with both Transformer-based and CNN-based generators and discriminators.

Table 2: **ViTGAN on CIFAR-10 when paired with CNN-based generators or discriminators**.

| Generator | Discriminator | FID ↓ | IS ↑ |
|---|---|---|---|
| ViTGAN | ViTGAN | 4.92 | 9.69 |
| StyleGAN2 | StyleGAN2 | 5.60 | 9.41 |
| StyleGAN2 | Vanilla-ViT | 17.3 | 8.57 |
| StyleGAN2 | ViTGAN | 8.02 | 9.15 |
| ViTGAN | StyleGAN2 | **4.57** | **9.89** |

**Generator architecture**  Table 3a shows GAN performances under three generator architectures, as shown in Fig. 2. Fig. 2 (B) underperforms other architectures. We find that Fig. 2 (A) works well but lags behind Fig. 2 (C) due to its instability. Regarding the mapping between patch embedding and pixels, implicit neural representation (denoted as NeurRep in Table 3a) offers consistently better performance than linear mapping, suggesting the importance of implicit neural representation in the ViTGAN generators.

**Discriminator regularization**  Table 3b validates the necessity of the techniques discussed in Section 4.1. First, we compare GAN performances under different regularization methods. Training GANs with ViT discriminator under R1 penalty (Mescheder et al., 2018) is highly unstable, as shown in Fig. 4, sometimes resulting in complete training failure (indicated as IS=NaN in Row 1 of Table 3b). Spectral normalization (SN) is better than R1 penalty. But SN still exhibits high-variance gradients and therefore suffers from low quality scores. Our $L_2$+ISN regularization improves the stability significantly (*c.f*.Fig. 4) and achieves the best IS and FID scores as a consequence. On the other hand, the overlapping patch is a simple trick that yields further improvement over the $L_2$+ISN method. However, the overlapping patch by itself does not work well (see a comparison between

Row 3 and 9). The above results validate the essential role of these techniques in achieving the final performance of the ViTGAN model.

Table 3: **Ablation studies of ViTGAN on CIFAR-10.**

| Embedding | Output Mapping | FID ↓ | IS ↑ |
|---|---|---|---|
| Fig 2 (A) | Linear | 14.3 | 8.60 |
| Fig 2 (A) | NeurRep | 11.3 | 9.05 |
| Fig 2 (B) | Linear | 328 | 1.01 |
| Fig 2 (B) | NeurRep | 285 | 2.46 |
| Fig 2 (C) | Linear | 15.1 | 8.58 |
| Fig 2 (C) | NeurRep | **6.66** | **9.30** |

(a) **Ablation studies of generator architectures.** NeurRep denotes implicit neural representation. ViT discriminator without convolutional projection is used for this ablation.

| Aug. | Reg. | Overlap | Proj. | FID ↓ | IS ↑ |
|---|---|---|---|---|---|
| ✗ | R1 | ✗ | ✗ | 2e4 | NaN |
| ✗ | R1 | ✓ | ✗ | 129 | 4.99 |
| ✓ | R1 | ✓ | ✗ | 13.1 | 8.71 |
| ✗ | SN | ✗ | ✗ | 121 | 4.28 |
| ✓ | SN | ✗ | ✗ | 10.2 | 8.78 |
| ✓ | $L_2$+SN | ✗ | ✗ | 168 | 2.36 |
| ✓ | ISN | ✗ | ✗ | 8.51 | 9.12 |
| ✓ | $L_2$+ISN | ✗ | ✗ | 8.36 | 9.02 |
| ✓ | $L_2$+ISN | ✓ | ✗ | **6.66** | **9.30** |
| ✓ | $L_2$+ISN | ✓ | ✓ | **4.92** | **9.69** |

(b) **Ablation studies of discrminator regularization.** 'Aug.', 'Reg.', 'Overlap' and 'Proj.' stand for DiffAug Zhao et al. (2020a) + bCR Zhao et al. (2021), and regularization method, overlapping image patches, and convolutional projection, respectively.

**Necessity of self-attention** Two intrinsic advantages of ViTs over CNNs are (a) adaptive connection weights based on both content and position, and (b) globally contextualized representation. We show the importance of each of them by ablation analysis in Table 4. We conduct this ablation study on the LSUN Bedroom dataset considering that its large-scale (∼3 million images) helps compare sufficiently trained vision transformers.

Table 4: **Ablations studies of self-attention on LSUN Bedroom 32x32.**

| Method | attention | local | global | FID ↓ |
|---|---|---|---|---|
| StyleGAN2 | ✗ | ✓ | ✗ | 2.59 |
| ViT | ✓ | ✗ | ✓ | 1.94 |
| MLP-Mixer | ✗ | ✗ | ✓ | 3.23 |
| ViT-local | ✓ | ✓ | ✗ | 1.79 |
| ViTGAN | ✓ | ✓ | ✓ | **1.57** |

First, to see the importance of adapting connection weights, we maintain the network topology (*i.e.*, each token is connected to all other tokens) and use fixed weights between tokens as in CNNs. We adopt the recently proposed MLP-Mixer (Tolstikhin et al., 2021) as an instantiation of this idea. By comparing the 2nd and 3rd rows of the table, we see that it is beneficial to adapt connection weight between tokens. The result of MLP-Mixer on the LSUN Bedroom dataset suggests that the deficiency of weight adaptation cannot be overcome by merely training on a large-scale dataset.

Second, to see the effect of self-attention scope (global *v.s.* local), we adopt a local self-attention of window size=3×3 (Ramachandran et al., 2019). The ViT based on local attention (4th row in Table 4) outperforms the ones with global attention. This shows the importance of considering local context (2nd row in Table 4). The ViT with convolutional projection (last row in Table 4) — which considers *both local and global contexts* — outperforms other methods.

For results on applying these techniques into either one of discriminator or generator, please refer to Table 6 in the Appendix.

## 6 CONCLUSION AND LIMITATIONS

We have introduced ViTGAN, leveraging Vision Transformers (ViTs) in GANs, and proposed essential techniques to ensuring its training stability and improving its convergence. Our experiments on standard benchmarks demonstrate that the presented model achieves comparable performance to leading CNN-based GANs. Regarding the limitations, ViTGAN is a novel GAN architecture consisting solely of Transformer layers. This could be improved by incorporating advanced training (*e.g.*, (Jeong & Shin, 2021; Schonfeld et al., 2020)) or coarse-to-fine architectures (*e.g.*, (Liu et al., 2021b)) into the ViTGAN framework. Besides, we have not verified ViTGAN on high-resolution images. Our paper establishes a concrete baseline of ViT on resolutions up to 256×256 and we hope that it can facilitate future research to use vision transformers for high-resolution image synthesis.

## ETHICS STATEMENT

We acknowledge that the image generation techniques have the risks to be misused to produce misleading information. Researchers should explore the techniques responsibly. Future research may have ethical and fairness implication especially when generating images of identifiable faces. It is unclear whether the proposed new architecture would differentially impact groups of people of certain appearance (*e.g.*, skin color), age (young or aged), or with a physical disability. These concerns ought to be considered carefully. We did not undertake any subject studies or conduct any data-collection for this project. We are committed in our work to abide by the eight General Ethical Principles listed at ICLR Code of Ethics (`https://iclr.cc/public/CodeOfEthics`)

## REPRODUCIBILITY STATEMENT

All of our experiments are conducted on public benchmarks. We describe the implementation details in the experiment section and in the Appendix C and Appendix D. We will release our code and models to ensure reproducibility.

## ACKNOWLEDGMENTS

This work was supported in part by Google Cloud Platform (GCP) Credit Award. We would also like to acknowledge Cloud TPU support from Google's TensorFlow Research Cloud (TRC) program. Also, we thank the anonymous reviewers for their helpful and constructive comments and suggestions.

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

# A    MORE RESULTS

## A.1    EFFECTS OF DATA AUGMENTATION

Table 5 presents the comparison of the Convolution-based GAN architectures (BigGAN and Style-GAN2) and our Transformer-based architecture (ViTGAN). This table complements the results in Table 1 of the main paper by a closer examination of the network architecture performance with and without using data augmentation. The differentiable data augmentation (DiffAug) (Zhao et al., 2020a) is used in this study.

As shown, data augmentation plays more critical role in ViTGAN. This is not unexpected because discriminators built on Transformer architectures are more capable of over-fitting or memorizing the data. DiffAug increases the diversity of the training data, thereby mitigating the overfitting issue in adversarial training. Nevertheless, with DiffAug, ViTGAN performs comparably to the leading-performing CNN-based GAN models: BigGAN and StyleGAN2.

In addition, Table 5 includes the model performance without using the balanced consistency regularization (bCR) (Zhao et al., 2021).

Table 5: Effectiveness of data augmentation and regularization on the CIFAR-10 dataset. ViTGAN results here are based on the ViT backbone without convolutional projection.

| Method | Data Augmentation | Conv | FID $\downarrow$ | IS $\uparrow$ |
|---|---|---|---|---|
| StyleGAN2 | None | Y | 8.32 | 9.21 |
| StyleGAN2 | DiffAug | Y | 5.60 | 9.41 |
| ViTGAN | None | N | 30.72 | 7.75 |
| ViTGAN | DiffAug | N | 6.66 | 9.30 |
| ViTGAN w/o. bCR | DiffAug | N | 8.84 | 9.02 |

## A.2    FID OF STYLEGAN2

We implemented StyleGAN2, Vanilla-ViT, and ViTGAN on the same codebase to allow for a fair comparison between the methods. Our implementation follows StyleGAN2 + DiffAug (Zhao et al., 2020a) and reproduces the StyleGAN2 FID reported in (Zhao et al., 2020a) which was published in NeurIPS 2020: the FID of our StyleGAN2 (+ DiffAug) re-implementation is 5.60 versus the 5.79 FID reported in Zhao et al. (2020a).

There is another contemporary work called StyleGAN2 (+ ADA) (Karras et al., 2020a). Both StyleGAN2 (+ DiffAug) (Zhao et al., 2020a) and StyleGAN2 (+ ADA) (Karras et al., 2020a) use the same StyleGAN2 architecture which is considered as the state-of-the-art CNN-based GAN. Due to their differences in data augmentation methods and hyperparameter settings, they report different FIDs on the CIFAR-10 dataset.

Note our goal is to compare the architecture difference under the same training setting, we only compare StyleGAN2 (+ DiffAug) (Zhao et al., 2020a) and use the same data augmentation method DiffAug in both StyleGAN2 and ViTGAN.

## A.3    NECESSITY OF SELF-ATTENTION

In Table 6, we extend the result from Table 4 by applying self-attention or local attention to either one of discriminator (D) or generator (G). The comparison between ViT vs. MLP shows that it is more important to use adaptive weights on the generator than the discriminator. The comparison between ViT and ViT-local shows that it is beneficial to have locality on both generator and discriminator.

## A.4    HIGH-RESOLUTION SAMPLES

Figure 5 and Figure 6 show uncurated samples from our ViTGAN model trained on LSUN Bedroom $128 \times 128$ and LSUN Bedroom $256 \times 256$, respectively. For $256 \times 256$ resolution, we use the

Table 6: **Detailed ablations studies of self-attention on LSUN Bedroom 32x32**.

| Method | FID $\downarrow$ |
|---|---|
| ViT-D + ViT-G | 1.94 |
| MLP-D + ViT-G | 2.33 |
| ViT-D + MLP-G | 2.86 |
| MLP-D + MLP-G | 3.23 |
| ViT-local-D + ViT-local-G | 1.79 |
| ViT-local-D + ViT-G | 2.03 |
| ViT-D + ViT-local-G | 2.04 |

same architecture as $128 \times 128$ resolution except for increasing the sequence length to 1024 for the generator. The FID of ViTGAN on LSUN Bedroom $256 \times 256$ is 4.67. Our method outperforms GANformer (Hudson & Zitnick, 2021), which achieved an FID of 6.51 on LSUN Bedroom $256 \times 256$.

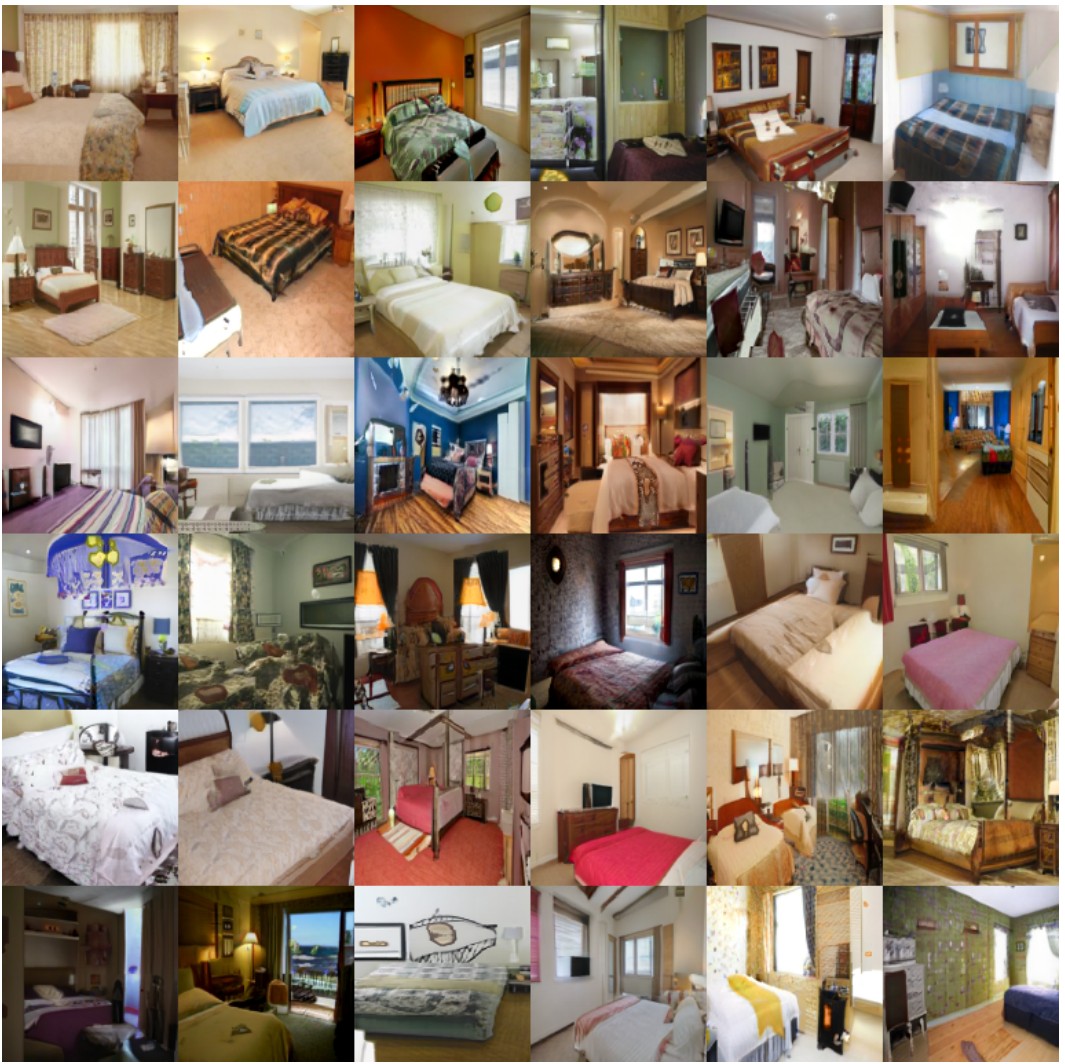

Figure 5: **Samples from ViTGAN on** $128 \times 128$ **resolution.** Our ViTGAN was trained on LSUN Bedroom $128 \times 128$ dataset. We achieved the FID of 2.48.

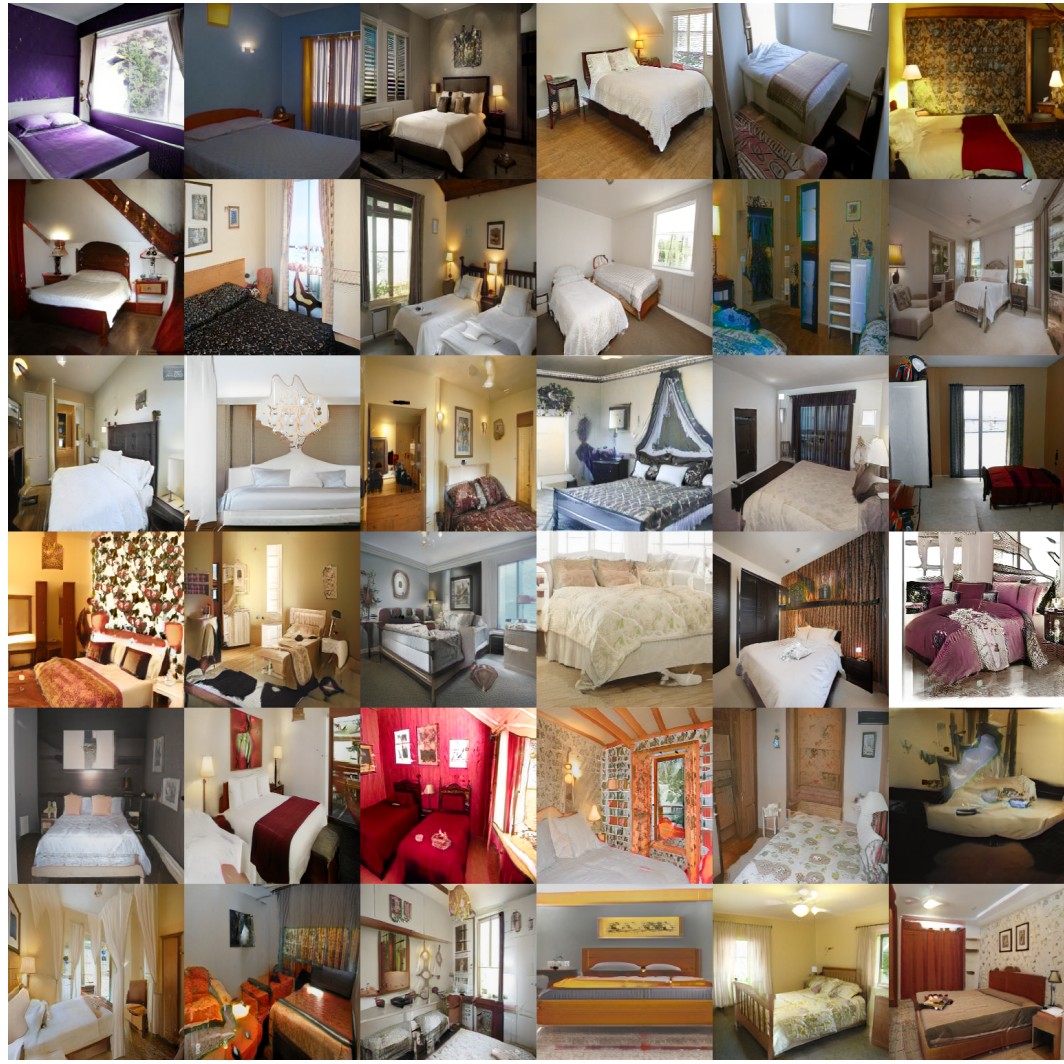

Figure 6: **Samples from ViTGAN on** $256 \times 256$ **resolution.** Our ViTGAN was trained on LSUN Bedroom $256 \times 256$ dataset. We achieved the FID of 4.67.

## B    COMPUTATION COST ANALYSIS

We would like to note that computational complexity is not the main challenge of using Transformers for high-resolution datasets. Due to various smart tokenization strategies such as non-overlapping patches (Dosovitskiy et al., 2021), stack of strided convolutions (Xiao et al., 2021), *etc*, we are able to suppress the explosion of effective sequence length — it is typically just a few hundred for standard image resolutions. As a result, the sequence length stays within a region where a quadratic time complexity is not a big issue. Actually, as shown in Table 7, ViTs are much more compute efficient than CNN (StyleGAN2). For our ViT-based discriminator, owing to the approach from (Xiao et al., 2021), we were able to scale up from $32\times32$ to $128\times128$ without the increase in sequence length (=64). For our ViT-based generator, since we did not employ any convolution stack, we had to increase the sequence length as we increased the resolution. This issue, in principle, can be addressed by adding up-convolutional (deconvolutional) blocks or Swin Transformer (Liu et al., 2021b) blocks after the initial low-resolution Transformer stage. Table 7 reveals that we are able to suppress the increase in computation cost at higher-resolutions by using Swin Transformer blocks. For this hypothetical ViTGAN-Swin variant, we use Swin Transformer blocks starting from $128^2$ resolution stage, and the number of channels is halved for every upsampling stage as done in Swin Transformer.

Please note that the main goal of this paper is not to design a new Transformer block for general computer vision applications; we instead focus on addressing challenges unique to the combination of Transformers and GANs. Techniques presented in this paper are still valid for numerous variants of ViTs such as Swin (Liu et al., 2021b), CvT (Wu et al., 2021), CMT (Guo et al., 2021), *etc*.

The experiments in the current paper is *not* based on Swin Transformers since using Swin Transformers resulted in inferior FID according to our experiments. However, we believe that with careful tuning of architectural configurations such as numbers of heads, channels, layers, *etc.*, ViTGAN-Swin may be able to achieve a better FID score. We were unable to perform this due to our limited computational resources. We leave leave it to future work.

Table 7: **Generator computation cost comparison among methods**.

| Method | #Params@$64^2$ | FLOPs@$64^2$ | FLOPs@$128^2$ | FLOPs@$256^2$ |
|---|---|---|---|---|
| StyleGAN2 (Karras et al., 2020b) | 24M | 7.8B | 11.5B | 15.2B |
| ViTGAN | 38M | 2.6B | 11.8B | 52.1B |
| ViTGAN-Swin | N/A | N/A | 3.1B | 3.5B |

## C    EXPERIMENT DETAILS

**Datasets**    The *CIFAR-10* dataset (Krizhevsky et al., 2009) is a standard benchmark for image generation, containing 50K training images and 10K test images. Inception score (IS) (Salimans et al., 2016) and Fréchet Inception Distance (FID) (Heusel et al., 2017) are computed over the 50K images. The *LSUN bedroom* dataset (Yu et al., 2015) is a large-scale image generation benchmark, consisting of ∼3 million training images and 300 images for validation. On this dataset, FID is computed against the training set due to the small validation set. The *CelebA* dataset (Liu et al., 2015) comprises 162,770 unlabeled face images and 19,962 test images. We use the *aligned* version of CelebA, which is different from *cropped* version used in prior literature (Radford et al., 2016; Jiang et al., 2021).

**Implementation Details.**    We consider three resolution settings: $32\times32$ on the CIFAR dataset, $64\times64$ on CelebA and LSUN bedroom, and $128\times128$ on LSUN bedroom. For $32\times32$ resolution, we use a 4-block ViT-based discriminator and a 4-block ViT-based generator. For $64\times64$ resolution, we increase the number of blocks to 6. Following ViT-Small (Dosovitskiy et al., 2021), the input/output feature dimension is 384 for all Transformer blocks, and the MLP hidden dimension is 1,536. Unlike (Dosovitskiy et al., 2021), we choose the number of attention heads to be 6. We find increasing the number of heads does not improve GAN training. For $32\times32$ resolution, we use patch size $4\times4$, yielding a sequence length of 64 patches. For $64\times64$ resolution, we simply increase the patch size to $8\times8$, keeping the same sequence length as in $32\times32$ resolution. For $128\times128$ resolution

generator, we use patch size $8 \times 8$ and 8 Transformer blocks. For $128 \times 128$ resolution discriminator, we maintain the sequence length of $64$ and 4 Transformer blocks. Similarly to (Xiao et al., 2021), $3 \times 3$ convolutions with stride $2$ were used until desired sequence length is reached.

Translation, Color, Cutout, and Scaling data augmentations (Zhao et al., 2020a; Karras et al., 2020a) are applied with probability $0.8$. All baseline transformer-based GAN models, including ours, use balanced consistency regularization (bCR) with $\lambda_{real} = \lambda_{fake} = 10.0$. Other than bCR, we do not employ regularization methods typically used for training ViTs (Touvron et al., 2020) such as Dropout, weight decay, or Stochastic Depth. We found that LeCam regularization (Tseng et al., 2021), similar to bCR, improves the performance. But for clearer ablation, we do not include the LeCam regularization. We train our models with Adam with $\beta_1 = 0.0$, $\beta_2 = 0.99$, and a learning rate of $0.002$ following the practice of (Karras et al., 2020b). In addition, we employ non-saturating logistic loss (Goodfellow et al., 2014), exponential moving average of generator weights (Karras et al., 2018), and equalized learning rate (Karras et al., 2018). We use a mini-batch size of $128$.

Both ViTGAN and StyleGAN2 are based on Tensorflow 2 implementation[2] and trained on Google Cloud TPU v2-32 and v3-8.

## D  IMPLEMENTATION NOTES

**Patch Extraction**    We use a simple trick to mitigate the over-fitting of the ViT-based discriminator by allowing some overlap between image patches. For each border edge of the patch, we extend it by $o$ pixels, making the effective patch size $(P + 2o) \times (P + 2o)$, where $o = \frac{P}{2}$. Although this operation has a connection to a convolution operation with kernel $(P + 2o) \times (P + 2o)$ and stride $P \times P$, we do not regard it as a convolution operator in our model because we do not use convolution in our implementation. Note that the extraction of (non-overlapping) patches in the Vanilla ViT (Dosovitskiy et al., 2021) also has a connection to a convolution operation with kernel $P \times P$ and stride $P \times P$.

**Positional Embedding**    Each positional embedding of ViT networks is a linear projection of patch position followed by a sine activation function. The patch positions are normalized to lie between $-1.0$ and $1.0$.

**Implicit Neural Representation for Patch Generation**    Each positional embedding is a linear projection of pixel coordinate followed by a sine activation function (hence the name Fourier encoding). The pixel coordinates for $P^2$ pixels are normalized to lie between $-1.0$ and $1.0$. The 2-layer MLP takes positional embedding $\mathbf{E}_{fou}$ as its input, and it is conditioned on patch embedding $\mathbf{y}^i$ via weight modulation as in (Karras et al., 2020b; Anokhin et al., 2021).

---

[2]`https://github.com/moono/stylegan2-tf-2.x`

