# OpenReview forum: "ViTGAN: Training GANs with Vision Transformers"
_ICLR.cc/2022/Conference — ICLR 2022 Spotlight_

### Official Review · Reviewer_mS8C · 2021-11-01

**Correctness:** 4
**Technical Novelty And Significance:** 2
**Empirical Novelty And Significance:** 3
**Recommendation:** 6
**Confidence:** 4

**Details Of Ethics Concerns:**

The proposed method itself does not lead to extra significant ethics concerns compared to existing GAN methods.

**Main Review:**

Pros:
1. The paper is generally well written and easy to follow.
2. The authors analyze the limitations of directly applying ViT models to training GANs and adopt several techniques to remedy such issues during the training.
3. Comprehensive ablation study shows the benefits of each component.

Cons:
1. During the evaluation, the authors only provide the FID numbers without providing the computation statistics (e.g., FLOPs) or model size. As we know, existing GAN models (e.g., StyleGAN2, StyleGAN3) can be scaled up to further improve the performance (especially when augmentation is used to prevent overfitting). Therefore, without providing the computation/model size of different models, it is hard to demonstrate if the proposed model actually has an advantage.
2. The proposed architecture seems to still be applicable to low-resolution images. It would be better to investigate the feasibility of scaling up to high-resolution images (e.g., 1024x1024). For example, showing the results on a small-scale dataset and measuring the computation on large resolutions would be great.
3. Many of the techniques like L2 attention to enforce Lipschitz constant (Kim et al., 2021), overlapping patches (Liu et al., 2021), convolutional projection (Wu et al., 2021; Guo et al., 2021), self-modulation (Chen et al., 2019), Fourier features (Tancik et al., 2020) are already proposed in existing work. The work seems more like a study combining existing techniques.
4. In Table 5, the FID of StyleGAN2 is worse with DiffAug, which is contrary to the results from (Zhao et al., 2021). Is there any reason why it happens?

**Summary Of The Paper:**

In this paper, the authors study employing ViTs for GAN-based image generation. For discriminators, they propose improved methods to enforce Lipschitz constraints and spectral norm regularization, and modified architecture; for generators, they adopted self-modulated LayerNorm and Fourier features to improve the performance. Experiments on CIFAR, LSUN bedroom, and CelebA show superior performance compared to existing methods.

**Summary Of The Review:**

The paper is overall well written and presents a solid study of using ViT models to build image GANs. However, the lack of computation/model size statistics may lead to unfair comparison; the novelty might be limited since most of the techniques are already proposed. I would like to hear from the authors for the final decision.

---

### Official Review · Reviewer_2CzQ · 2021-11-01

**Correctness:** 4
**Technical Novelty And Significance:** 2
**Empirical Novelty And Significance:** 3
**Recommendation:** 6
**Confidence:** 4

**Main Review:**

The paper is very well written and structured. I like the connection of recent results in transformer literature regarding Lipschitzness of self-attention and its direct implication and application in the GAN setting. The ablation studies of the proposed changes are mainly convincing, the proposed changes all contribute to the final performance.

There are several major issues that the authors should address:
- **Usefulness of ISN**: It is not clear why R1 penalty works so much worse for ViTs; I would expect the regularization to be independent of the network architecture. The authors write that “our method overcomes the spikes in gradient magnitude and stabilizes the training dynamics” - R1 seems to be the best at least regarding gradient magnitude; why does collapse happen then? Any intuitions or further investigation would be very interesting. Also, what is the gamma value for R1? Maybe I missed it, but I could not find it in the paper. This detail is crucial, as gamma is dependent on the dataset, and it is necessary to do sweeps for good values. A good gamma value might make one of the main contributions (ISN) less important. Of course, it is still good not having to do parameter sweeps, which would be an advantage of ISN.
- **Baselines**: the code for TransGAN is available and it is the primary ViT-based baseline. Hence, it would be important to see how well TransGAN does on other datasets except for CiFAR when compared to ViTGAN. Furthermore, the StyleGAN2 baseline on CIFAR should include data augmentation, which would improve over ViTGAN (Stylegan2-ADA: FID=2.9). I don’t mind the numbers not being bold for the proposed approach; instead, numbers from the literature should be reported faithfully.
-**Limitations**: This is not a technical issue, but it should clearly state why it is hard to scale ViTGAN up to higher resolutions. Being more upfront about the problem in the limitation section would be nice. The authors aim to close the gap to the current state-of-the-art GAN, but GANs can operate on >1MP resolution. The authors should expand more on the challenges of scaling up such models. On a side note, TransGAN did evaluate on 256x256, so it is not impossible to scale (but challenging).
-**Contributions**: The main idea of using ViTs for GANs is not original itself, and has been done before (TransGAN). In terms of technical contributions, as mentioned above, I am still not convinced of the usefulness of ISN. Lastly, for an empirical study, the experiments are somewhat lacking and I would like to see more and stronger baselines and more datasets.

Minor Issues:

- **Quantitative samples**: If not stated otherwise, I assume the shown samples are cherrypicked? If yes, the authors should mention this in the figure caption. Also, normally such samples are done via truncation, is truncation possible with a ViTGAN?
- **General Motivation**: It is still unclear what the advantage would be to use a ViT in a GAN instead of standard CNNs (other than it just being generally interesting). The experiments regarding self-attention and adaptive weights are not very convincing, especially since a CNN discriminator improves over a ViT discriminator
- Last paragraph intro: “Stylegan2-ADA [...] is equipped with sophisticated data augmentation methods”, this is not an advantage of CNN GANs; the authors even run experiments with DiffAugment in the appendix.
- The authors mention SWIN transformers as a possible way to improve performance. Why don’t they run this experiment? This experiment should be a simple drop-in and also has been done before (TransGAN).
- Section 5 “necessity of self-attention”. The authors evaluate the usefulness of adaptive weights and global representations of ViTs. It is unclear if the experiments look at G, D, or both. If it is either one, it might be interesting to look a the other side as well.
- The experiments show that spectral norm at initialization (ISN) improves results. Why does it make sense to use the initial spectral norm? Currently, the motivation or intuition why it makes sense is missing.
- Does A correspond to style codes like the W space in a StyleGAN? This correspondence would be interesting to investigate. I suspect it will not be similar to style codes because of the non-hierarchical nature of a ViT and would be a minor downside of the current architecture.
- SLN is used ins Eq (9,10) but introduced later in the text

**Summary Of The Paper:**

This paper investigates the use of ViTs in GANs.  Currently, CNNs are the standard neural network architecture for GANs, and it is a consequential idea to apply the recent ViT architecture in the generative setting. A question the authors aim to answer is how standard regularization techniques like spectral normalization work for this different type of architecture since it is unclear if they are readily applicable. The authors propose an adaptation of spectral normalization that works better when using ViTs, and two modifications to the ViT generator.

**Summary Of The Review:**

This work is an empirical study and of interest to the community. The novelty of the idea and the technical contributions are somewhat limited.  As for empirical insights, there are several interesting findings, but one of the main contributions (ISN) needs to be compared to a stronger baseline, ie., a sweep over gamma values. Further, the comparisons to other baselines are currently incomplete, and the limitations will need more discussion. I rate this paper at 5, but I am happy to move up the score once the raised issues are addressed.

---

### Official Review · Reviewer_Wrur · 2021-11-02

**Correctness:** 3
**Technical Novelty And Significance:** 3
**Empirical Novelty And Significance:** 3
**Recommendation:** 6
**Confidence:** 4

**Main Review:**

***Strengths***:
1. The paper is presented clearly, particularly the method.

2. Most experiments can verify the effectiveness of the core components of the method.

***Weaknesses***:

Concern on the motivation:

1. What is the advantage of transformer over CNN in generative modeling? The authors claim to design a new GAN structure, but it is not clear what the advantages of the new structure are.

Concern on the experiments:

2. The paper chooses StyleGAN2 as a strong baseline. But I cannot find any comparison between the FLOPs and the parameters between ViTGAN and SyleganGAN2.

3. The last two rows of Table 1 demonstrate that the ViT-based discriminator is not very effective, so what is the advantage of the ViT-based discriminator? Besides, could the authors provide the result of StyleGAN2-G + ViTGAN-D? With the same discriminator, does ViTGAN-G work better than StyleGAN2-G?

4. Section 4.1 has presented the local and global text fusion is necessary when designing the discriminator. Swin transformer works with the same mechanism. Why don't the authors compare your design with Swin transformer?

5. The state-of-the-art GANs have advanced high-resolution image synthesis. Why not compare ViTGAN with these state-of-the-art GANs on high-resolution image synthesis？ Does the computational overhead make ViTGAN impossible to train on high-resolution images?


**Summary Of The Paper:**

This paper has presented a new GAN architecture based on the ViT.  Trivially extending ViT for image generation brings huge performance drop as well as unstable training. The authors have found that high-variance gradients in the transformer-based discriminator cause unstable training.  The new regularizers are proposed to stabilize the training. To better introduce the stochastics and leverage the local context, modifications on the generator such as self-modulated LayerNorm and the implicit functions of the path are facilitated for the training. The authors have conducted ViTGAN on several low-resolution datasets, and ablation studies are conducted to support the effectiveness of the main components of ViTGAN.

**Summary Of The Review:**

Concerns about the method and experiment results have been presented in Weaknesses.
I think the authors need more important experiments to support the advantages of ViT-based GAN over CNN-based GAN.

---

### Official Review · Reviewer_oHXw · 2021-11-02

**Correctness:** 4
**Technical Novelty And Significance:** 3
**Empirical Novelty And Significance:** 4
**Recommendation:** 8
**Confidence:** 5

**Details Of Ethics Concerns:**

none of Ethics Concerns.

**Main Review:**

Pros:
1. The architecture of the proposed method is quite clean and simple. The design of the ViT architecture part don't change much, compared with the original one. This paper mainly focus on finding regularization techniques and minor changes, which works for ViT.
2. Several useful techniques and tricks are proposed with detailed experiments, which could be referred and taken by other ViT+GAN works.
3. The paper is well-organized. Also, it contains a section of preliminaries for Vision Transformer, which might help new ViT+X follower.

Cons:
1. This paper might sound a bit lack of novelty to me, which looks like one of pursuers for ViT+X. It think the main tackle for making naive ViT not worth and meaningful to combine with GAN is the unacceptable and unaccessible computation cost and memory cost, when we're targeting on typical resolution 256x256, 512x512, even more.
2. With Cons#1. Basically, the authors take a less meaningful resolution baseline of range from 32x32 to 64x64, and one 128x128 for LSUN. I suggest experiments of resolution 256 or 512 would be helpful to provide.
3. Also, other computation cost related information could be a great plus, such as FLOPs or Latency(ms), which could make a more fair comparison with traditional CNN-based methods or other ViT+GAN methods.
4. Some of proposed techniques are well-introducted and involved by recent works.
1) Overlapping Image Patches:

FuseFormer: Fusing Fine-Grained Information in Transformers for Video Inpainting (ICCV 2021)

Swin Transformer: Hierarchical Vision Transformer using Shifted Windows (ICCV 2021)

2) Convolutional Projection:

CvT: Introducing Convolutions to Vision Transformers

**Summary Of The Paper:**

This paper introduces a novel new GAN design, named ViTGAN, which integrates the Vision Transformer(ViT) architecture into Generative Adversarial Networks(GAN), replace both discriminators and generators. Several novel regularization techniques are proposed for addressing the traditional regularization methods work poorly with self-attention by direct combination in discriminators. Also, a few of architectural choices are examined for latent and pixel mapping layers to faciliate convergence in generators. Combining all the empirical design and choices, the method ViTGAN achieves comparable performance to the SOTA CNN-based GAN models on mainstream metrics.

**Summary Of The Review:**

I recommend the paper would be worthy with the rating 6.
If all my concerns could be well-addressed, a score of 8 is deserved. And this works would be one of best ViT+GAN proposal.

---

### Official Review · Reviewer_vpzj · 2021-11-03

**Correctness:** 3
**Technical Novelty And Significance:** 3
**Empirical Novelty And Significance:** 2
**Recommendation:** 6
**Confidence:** 4

**Details Of Ethics Concerns:**

The authors have provided discussion to address underlying ethics concerns.

**Main Review:**

Strengths:

The authors give a comprehensive review and discussion to modern related works, including some contemporary approaches.

The whole paper is well organized and written, and easy to follow.

The direction of exploring transformer on image generation is novel, although several works have investigated Vision-Transformer and achieved competitive results on image recognition.

Detailed strategies are proposed for the components in transformer-based generators and discriminators. The work presents comprehensive ablation studies and promising results compared to CNN-based benchmarks.

Weaknesses:

There are several questions about the insight and effect of some design:
1. L2 attention is adopt in ViT-based discriminator duo to the consideration of training stability (i.e., Lipschitzness). How does it perform if using in the generator?
2. The capacity of discriminator has shown to be important to guiding the training of a high-performing generators (e.g., StyleGANs and BigGAN). The results in Table 2 also show that using a powerful CNN-based discriminator performs better than Transformer-based discriminator. Could the authors provide more insight or discussion about it? Locality that CNNs intrinsically have is supposed to be very important for vision tasks and whether it is necessary to eliminate such inductive biases.
3. The authors also point out the limitation of current method on high-resolution image generation.  The challenges of extending transformer to such a scenario may arise from several aspects, including computational overhead and effectiveness of using global context, as the low-level features in high-resolution is supposed to be naturally local.  It would be helpful if the authors could provide deeper insight or discussion about it.

Minor suggestions:  L is used to denote patch length in the section of ViT definition. It is a bit confusing to use L as well in Eqn. 2-4, which is supposed to represent layer index.


**Summary Of The Paper:**

The authors aim to investigate the effect of transformer, a convolution-free network architecture which has achieved impressive performance on NLP, on image generation. Vision Transformer works regard an image as a sequence of tokens The work presents several challenges of developing a vision-transformer encoder, including architecture designs for embedding latent vectors and patch generation in generators and some regularizations for stabilizing the training of VIT-based discriminators. The experiments show the proposed framework can achieve competitive performance with CNN-based GANs on several datasets with image resolutions up to 128x128.

**Summary Of The Review:**

The motivation of investigating the effect of modern architecture Transformer, without using spatial convolution or pooling, on image generation is interesting and valuable. Transformer-like architectures have shown remarkable performance on a wide range of tasks, including image recognition, while exploring it on GANs for image generation is a novel and challenging direction. The current framework (without significant modification to the original ViT) has shown promising results compared to CNN benchmarks. On the other hand, the paper can to be improved by providing more discussion and analysis about the relation between locality that convolution intrinsically has and global context that self-attention captures.

---

### Public Comment · ~Chaman_Ambedkar1 · 2022-01-30
**Rebuttal?**

The model has problems in scaling to high-resolution images (which was addressed in TransGAN, probably a major reason for the worsen performance with grid attention. 128 is a very low resolution nowadays), and the paper did not show any of #params / #FLOPs / #GPUs / training time that are crucial in comparing generative model performance. The reviewers / ACs seem to be more than satisfied with the rebuttal (scores from 55566 to 66686), but the rebuttal is somehow not publicly available and the paper did not make any related revision as well. Could the authors make their rebuttal public?

---

> ### Public Comment · ~Kwonjoon_Lee1 · 2022-01-31
> **Thank you**
>
> Thank you for your interest in our work. We will upload the revised paper along with the code as soon as possible.

---

### Decision · Program_Chairs · 2022-01-20

**Decision:**

Accept (Spotlight)

**Comment:**

The paper proposes a GAN architecture with a ViT-based discriminator and a ViT-based generator. The paper initially received a mixed rating with two "slightly above the acceptance threshold" ratings and "three slightly below the acceptance threshold" ratings. Several concerns were raised in the reviews, including whether there are advantages of using a ViT-based GAN architecture over the CNN-based GAN and whether the proposed method can be extended to high-resolution image synthesis. These concerns are well-addressed in the rebuttal with most of the reviewers increasing their ratings to be above the bar. The meta-reviewer agrees with the reviewers' assessments and would like to recommend acceptance of the paper.